# An evaluation of the evidence submitted to Australian alcohol advertising policy consultations

**Julia Stafford** [1,2]*, **Tanya Chikritzhs**[1], **Hannah Pierce**[2], **Simone Pettigrew**[1,3]

**1** National Drug Research Institute, Health Sciences, Curtin University, Perth, Western Australia, Australia, **2** Cancer Council Western Australia, Perth, Western Australia, Australia, **3** The George Institute for Global Health, University of New South Wales, Sydney, New South Wales, Australia

* julia.stafford@postgrad.curtin.edu.au

## Abstract

**Data Availability Statement:** The data underlying the results presented in the study are available from the Parliament of New South Wales Portfolio Committee No. 1 – Premier and Finance (website:

### Background

Industry self-regulation is the dominant approach to managing alcohol advertising in Australia and many other countries. There is a need to explore the barriers to government adoption of more effective regulatory approaches. This study examined relevance and quality features of evidence cited by industry and non-industry actors in their submissions to Australian alcohol advertising policy consultations.

### Methods

Submissions to two public consultations with a primary focus on alcohol advertising policy were analysed. Submissions (n = 71) were classified into their actor type (industry or non-industry) and according to their expressed support for, or opposition to, increased regulation of alcohol advertising. Details of cited evidence were extracted and coded against a framework adapted from previous research (primary codes: subject matter relevance, type of publication, time since publication, and independence from industry). Evidence was also classified as featuring indicators of higher quality if it was either published in a peer-reviewed journal or academic source, published within 10 years of the consultation, and/or had no apparent industry connection.

### Results

Almost two-thirds of submissions were from industry actors (n = 45 submissions from alcohol, advertising, or sporting industries). With few exceptions, industry actor submissions opposed increased regulation of alcohol advertising and non-industry actor submissions supported increased regulation. Industry actors cited substantially less evidence than non-industry actors, both per submission and in total. Only 27% of evidence cited by industry actors was highly relevant and featured at least two indicators of higher quality compared to 58% of evidence cited by non-industry actors.

https://www.parliament.nsw.gov.au/committees/listofcommittees/Pages/committees.aspx) and Figshare (website: https://figshare.com/articles/journal_contribution/Submissions_to_the_Alcohol_Advertising_Issues_Paper/4530581/1; DOI: https://doi.org/10.6084/m9.figshare.4530581.v1).

**Funding:** This work was supported by an Australian Government Research Training Program Scholarship. The Scholarship is provided by the Commonwealth of Australia to support general living costs for students (JS) undertaking Research Doctorate studies. The Scholarship provider had no role in the design of the study; in the collection, analyses, or interpretation of data; in the writing of the manuscript, and in the decision to publish the results. The research did not receive any specific grant from funding agencies in the public, commercial, or not-for-profit sectors.

**Competing interests:** I have read the journal's policy and the authors of this manuscript have the following competing interests: JS and HP contributed to the written submission of the McCusker Centre for Action on Alcohol and Youth to the inquiry into the NSW Alcoholic Beverages Advertising Prohibition Bill 2015 and JS appeared as a witness to a public hearing held for the inquiry. JS and HP contributed to the written submission of the McCusker Centre for Action on Alcohol and Youth to the ANPHA issues paper. TC was a member of the Australian National Preventive Health Agency's expert committee on alcohol and contributed to drafting of the final report.

## Conclusions

Evaluation of the value of the evidentiary contribution of industry actors to consultations on alcohol advertising policy appears to be limited. Modifications to consultation processes, such as exclusion of industry actors, quality requirements for submitted evidence, minimum standards for referencing evidence, and requirements to declare potential conflicts, may improve the public health outcomes of policy consultations.

## Introduction

The sustained burden of alcohol-caused harm has required governments to consider regulatory options for reducing levels of harm and associated public costs [1, 2]. Stakeholder consultation is a common part of government policy development processes [3, 4], including for alcohol policy [5–8]. Public consultation processes typically involve a general call for submissions [4], providing an opportunity for stakeholders to connect decision-makers with evidence and shape how the policy matter is perceived. In general, few requirements or conditions are placed on submissions to public consultations in terms of who may submit them, how they are presented, or the veracity of their content [6, 9, 10]. Concerns about the participation of alcohol industry actors in public health policy consultations have been raised by health and academic stakeholders due to the industry's conflicts of interest and documented behaviours of distorting and denying evidence [5, 7, 8, 11–15].

Where submissions are publicly accessible, they can provide a data source for evaluating the evidentiary contribution of different actors to policy consultations. To date, there has been limited work of this nature in the context of alcohol policy. In a study of alcohol industry submissions to a Scottish Parliamentary consultation on minimum unit pricing, Cullen et al. found that peer-reviewed evidence (i.e., evidence appraised to be of sufficient quality by independent academic reviewers) accounted for only 15% of 115 items of evidence cited across 25 industry submissions [5]. Cited evidence was most often drawn from government reports or data (45%) or private/non-government organisations (30%). Several evidence items cited in the industry submissions were published more than a decade before the consultation, and Cullen et al. noted that such dated texts could not provide up-to-date support for the submitters' claims in this context [5].

In New Zealand, the Ministerial Forum on Alcohol Advertising and Sponsorship was established in 2014 to examine whether additional controls on alcohol advertising and sponsorship were needed to reduce harm from alcohol. The Ministerial Forum held a public consultation process with a request for submitters to focus on evidence that had emerged since 2010. A commissioned analysis of submissions to the Ministerial Forum found that a large body of post-2010 evidence was presented by submitters who supported further restrictions on alcohol advertising, with these submitters most commonly from the health sector, community groups, local governments, and universities [16]. In contrast, a smaller body of post-2010 evidence was presented by submitters who opposed further restrictions, most commonly sporting bodies, advertising and media organisations, alcohol industry bodies, and retailers. Similar quantities of pre-2010 evidence were presented by submitters who supported and opposed further alcohol advertising restrictions.

More detailed assessments of evidence cited in industry actor submissions to government consultations can be found in the tobacco policy context. Hatchard, Fooks, Evans-Reeves, Ulucanlar, and Gilmore [17] and Evans-Reeves, Hatchard, and Gilmore [18] analysed tobacco

industry submissions to a UK consultation on standardised packaging in terms of the quality and relevance of cited evidence. Evidence was judged to be higher quality if it was peer-reviewed and independent of industry. Hatchard et al. found that most evidence cited by tobacco companies lacked either policy relevance (i.e., it was not focused on standardised packaging or tobacco packaging) or indicators of quality [17]. Evidence classified as relevant to the issue under consideration was often connected to industry and not peer reviewed [17]. Industry connections to cited evidence were not disclosed in submissions [18]. Evidence cited in submissions by ultra-processed food companies has similarly been found to lack independence from industry and other indicators of quality [19]. Researchers have recognised parallels in evidence-related tactics by different corporate sectors [19–21] and noted the potential value of applying similar analyses to other public health policy areas where corporate interests seek to influence policy, such as alcohol [17].

The focus on alcohol advertising in the present study was chosen for several reasons. First, a substantial body of research evidence is available to inform deliberations about alcohol advertising policy (for recent reviews see [22–26]). The evidence base has developed to a level where it can be concluded that exposure to alcohol advertising is a cause of alcohol use among young people [27, 28]. Second, limited policy progress towards mitigating the effects of alcohol advertising has been achieved [1, 29]. Governments in Australia, as in many countries, continue to rely on industry-led self-regulation of alcohol advertising despite the documented weaknesses of that approach [25, 30–34]. Third, achieving regulatory change to reduce exposure to alcohol advertising remains a priority goal of health stakeholders [35–38]. Exploring the barriers to governments adopting more effective regulatory approaches may provide insights into opportunities that facilitate policy progress.

Investigations by researchers of alcohol industry efforts to influence advertising policy have sought to identify industry actor strategies and common arguments made [15, 39]. The most frequently documented strategies were: (i) policy substitution, including via promotion of voluntary codes and other ineffective initiatives; (ii) constituency building via development of trade associations, mobilisation of allied industries, and formation of front groups; and (iii) information-related tactics, including examples of selective citation of favourable evidence, omission of evidence, and contesting evidence used to support policy [15]. Frames and arguments used by alcohol industry actors when opposing increased regulation closely mirror those used by the tobacco industry [15, 39, 40]. Common industry claims include that alcohol brands are marketed responsibly and only to adults in compliance with self-regulatory codes, making increased regulation unnecessary. Claims of adverse unintended consequences of regulation are also common and often relate to the cost of compliance, job losses in associated industries, and discouragement of healthier product development [8]. Industry actors have also argued that the evidence base is insufficient to support proposals to regulate further, as advertising is simply intended to inform consumer brand selection and influence market share [8, 15, 39].

The present study examined a range of relevance and quality features of evidence cited by industry actors in their submissions to alcohol advertising policy consultations and compared them with evidence cited by non-industry actors. The study aimed to address two key questions: (i) What evidence do industry and non-industry actors cite in submissions to alcohol advertising policy consultations in terms of subject matter relevance, publication type, time since publication, and independence from industry funding sources? and (ii) What proportions of the cited evidence feature indicators of higher quality and relevance? Implications were considered for strengthening the use of evidence in policy processes to improve public health outcomes.

## Materials and methods

### Design

A comparative analysis was conducted of the evidence cited in submissions to Australian policy consultations about alcohol advertising. Using a quantitative approach informed by Hatchard et al. and Evans-Reeves et al.'s analyses of tobacco industry submissions, relevance and quality features of evidence cited in submissions were assessed [17, 18].

### Data

Analysed data were submissions made to the two public consultations from the past 10 years that primarily focused on alcohol advertising policy. The consultations were undertaken by the Australian Federal Government (the Australian National Preventive Health Agency (ANPHA) [41]) and the New South Wales (NSW) State Parliament [42]. The ANPHA consultation considered the effectiveness of existing national regulations in addressing community concerns about alcohol advertising (n = 34 submissions responded to ANPHA's issues paper) and the NSW Parliament's 2017 consultation considered a Bill (the NSW Alcoholic Beverages Advertising Prohibition Bill) that proposed to reduce alcohol advertising within the state (n = 42 submissions). Submissions were publicly available from the relevant government or parliamentary websites, except for four confidential submissions that were not released (one was later released by the submitter and included in the analysis). Two submissions to the ANPHA consultation were excluded (one was only a cover letter and another appeared to be incorrectly listed as a submission). In total, 71 submissions (n = 30 from the ANPHA consultation and n = 41 from the NSW consultation) were included in the analysis.

### Procedure for submission-level variables

Page lengths of the main bodies of submissions and appendices (if present) and the number of evidence items cited in each submission were counted to describe the composition of submissions made to the two consultations. Each submission was coded by type of actor (industry or non-industry) and expressed position on increased regulation of alcohol advertising (support, neutral, or oppose). Industry actors included alcohol trade associations, alcohol producers and retailers, advertising and media organisations, and sporting organisations with commercial relationships with members of the alcohol industry (such as via sponsorship). All other submissions were classified as originating from non-industry actors. Submissions were classified as supportive of increased regulation of alcohol advertising if they contained at least one statement expressing support for increased regulation. Conversely, submissions were classified as opposing increased regulation of alcohol advertising if they contained at least one statement expressing opposition to increased regulation or government regulation of alcohol advertising. Where no clear indication of position was provided either way, submissions were classified as neutral.

### Procedure for coding evidential criteria

For each submission, one researcher (JS) extracted details of cited evidence into a spreadsheet (bibliographic entry was copied and pasted without editing). If an item of evidence was cited more than once within a submission, it was only recorded once in the spreadsheet. To be extracted, sufficient information on the source of cited evidence was required. An appendix to a submission (e.g., a report or journal publication) was treated as an item of evidence and its details (author, title, date) were recorded. Attempts were made to access all cited evidence via the internet (using reasonable minimal effort) and the outcome recorded. The Wayback

**Table 1. Coding framework for classifying evidence.**

| | Evidential criteria | | Coding categories |
|---|---|---|---|
| Relevance | Subject matter | What is the topic of the evidence? | Alcohol advertising (high relevance) |
| | | | Alcohol but not advertising (medium relevance) |
| | | | Advertising but not alcohol (medium relevance) |
| | | | Unrelated to alcohol or advertising (low relevance) |
| | | | Unknown |
| Quality | Publication type | Where was the evidence published? | Peer-reviewed journal[a] |
| | | | Academic publication (e.g., book, university-published report, conference abstract)[a] |
| | | | Government publication[b] |
| | | | Parliamentary publication[b] |
| | | | Publication by a company or organisation not linked to the alcohol industry[b] |
| | | | Media coverage[b] |
| | | | Publication by an alcohol industry-linked company or organisation |
| | | | Publication from an industry self-regulatory body |
| | | | Unknown |
| | Time since publication | When was the evidence published? | Published within 10 years of the consultation[c] |
| | | | Published more than 10 years before the consultation |
| | | | Unknown |
| | Independence | Does the peer-reviewed journal or academic publication have any connection with the alcohol industry? | Independent of the alcohol industry[b] |
| | | | No apparent alcohol industry connection[b] |
| | | | Alcohol industry-funded |
| | | | Alcohol industry-linked |

[a]This coding category forms part of an indicator of higher quality: Published in a peer-reviewed journal or academic source.

[b]This coding category forms part of an indicator of higher quality: Independent of or with no apparent connection to the alcohol industry.

[c]This coding category is an indicator of higher quality: Published within 10 years of the consultation.

Machine online archive was used to access website sources as they existed at the date of citation [43].

Each item of evidence was classified against a four-part framework (see Table 1) adapted from Hatchard et al. and Evans-Reeves et al.'s analyses of tobacco industry submissions [17, 18]. In some instances, cited evidence was not accessible but bibliographic entries within the submission contained information that could be classified against the framework (e.g., included a title that clearly indicated the subject matter or year of publication). Where the necessary information was not accessible, items were coded as unknown.

**Subject matter relevance.** The subject matter of evidence cited in submissions was evaluated in terms of relevance to the issue under consideration in the consultation (i.e., alcohol advertising). Subject matter was assessed based on the presence or absence of key terms related to alcohol (alcohol, liquor, beer, wine, spirits, brand names) and advertising (advert*, market*, promot*, sponsor*) in the publication. High relevance was coded for publications in which one or more alcohol <u>and</u> one or more advertising terms were present. Medium relevance was coded where either one or more alcohol <u>or</u> one or more advertising terms were present. The low relevance code applied to publications where <u>neither</u> alcohol nor advertising terms were identified.

**Publication type.** Publication type was coded based on the category that best described where evidence had been published (see Table 1 for the nine coding categories). If a report

indicated that it had been commissioned (e.g., by a government department), publication type was determined based on the nature of the commissioning body (e.g., official government publication). Publications produced by an alcohol industry-linked company or organisation included those from: (i) alcohol producers, (ii) alcohol industry trade associations and news services (e.g., The Shout), (iii) social aspects organisations (e.g., DrinkWise), (iv) advertising and media agencies and associations that are likely to have alcohol industry clients (e.g., Google, Facebook), and (v) consulting or research/data organisations where the business community is a key customer (e.g., Deloitte, Access Economics, KPMG, Roy Morgan, IBISWorld, Nielsen, Oztam). Media coverage included general news outlets and advertising industry news services (e.g., Mumbrella, Ad Age).

**Time since publication.**    Time since publication (i.e., age of evidence) was identified by Cullen et al. [5] as important to evidence assessment and was included in this study to account for substantial changes in the advertising landscape in terms of technologies available to advertisers and structure of the global alcohol production industry [29]. Evidence was coded as published within 10 years of the consultation based on publication year (i.e., ≥2003 for evidence cited in submissions to ANPHA and ≥2008 for evidence cited in submissions to NSW Parliament).

**Independence.**    Peer-reviewed journal and academic publications were coded for independence from industry. Information provided within the publication (e.g., funding acknowledgement or conflict declaration) was used to inform the coding decision. Where the coder was unfamiliar with the funder, the funder's website was searched for evidence of industry connections. To be coded as independent, the publication needed to either: (i) name funding sources that were not connected to alcohol or advertising industries or (ii) declare that there were no conflicts. A publication that declared a funding source within the alcohol industry was coded as industry-funded. To be coded as industry-linked evidence, the publication needed to declare a connection between an author or the publication and either: (i) the alcohol industry other than as a funder of the published work or (ii) the advertising industry. If insufficient information was provided to allow for one of the above classifications, independence was coded as having no apparent alcohol industry connection.

**Indicators of higher quality evidence.**    As shown in Table 1, indicators of higher quality included whether evidence was:

1. Published in a peer-reviewed journal or academic source (e.g., book, conference abstract). Peer-reviewed journal publications have been assessed by external researchers knowledgeable in the subject area, acting as a filter to prevent poorly designed studies from being published [17];

2. Independent of or with no apparent connection to the alcohol industry. Connections with financially vested interests can cast doubt on the independence and quality of evidence due to the potential for bias associated with industry influence [17, 44]; and/or

3. Published within 10 years of the consultation. More recent work is more likely to reflect the contemporary advertising landscape [5, 29, 45] and improvements in research methodologies [46–48].

## Inter-coder reliability

Sub-samples (25%) from submissions (n = 18) and cited evidence (n = 307) were randomly selected and individually coded by two researchers (JS, HP) to gauge inter-coder reliability of the assessment of submission-level and evidence-level variables. There was 100% agreement

on coding of submission-level variables (actor type and position on increased regulation of alcohol advertising). Coding of evidence-level variables against the four-part framework resulted in 84 coding differences across the 1228 coded cells (6.8%). Differences were discussed and resolved between the two coders. The remaining submissions and cited evidence were coded by JS.

## Statistical analysis

Frequencies were calculated for the submission-level variables. For the evidence-level variables, frequencies across the four-part framework were calculated separately for evidence cited by industry and non-industry actors. Frequencies of evidence that featured any two or all three indicators of higher quality were calculated. Chi-square tests were used to assess differences in the proportions of evidence cited by industry and non-industry actors, with a Bonferroni correction to account for the number of comparisons (0.05/19 = 0.0026). Data were eyeballed for potential outlier scores. Where detected, analyses were re-run with outlier scores removed to determine whether results differed.

## Results

### Submission characteristics

The majority of submissions were made by industry actors (63%, n = 45, Table 2), of which half (51%, n = 23) were from advertising and media organisations (S1 Table). All but one of the industry actor submissions (44 of 45) opposed increased regulation of alcohol advertising (one was neutral). Conversely, 25 of the 26 non-industry actor submissions expressed support for increased regulation of alcohol advertising (one was neutral).

Submission size varied substantially within and between actor groups in terms of page length and quantity of evidence cited (Table 2). A quarter (24%, n = 11) of industry actor submissions did not cite any evidence; non-industry actor submissions each cited at least two items of evidence. In total, 520 items of evidence were cited by industry actors (mean of 12 items per submission) and 730 items of evidence were cited by non-industry actors (mean of 28 items per submission).

### Comparison of evidence cited by industry and non-industry actors

Not all evidence was able to be accessed; 14% (73 of 520) of evidence items cited by industry actors and 6% (44 of 730) of evidence items cited by non-industry actors were not accessible (see S2 Table). Publications by alcohol industry-linked organisations were the least accessible of the publication types cited by industry actors (37 of 73); for non-industry actors, government publications were the least accessible (15 of 44).

Table 3 summarises results for comparisons of proportions between industry and non-industry actor submissions for the four key measures relating to evidence cited. Industry actors cited a significantly lower proportion of evidence that was highly relevant (i.e., about alcohol advertising; 50% vs 73%, $p < .0001$) and a significantly higher proportion of evidence of medium relevance (i.e., about alcohol or advertising but not alcohol advertising; 37% vs 22%, $p < .0001$) than non-industry actors.

Peer-reviewed journal publications were the most cited publication type for both actor groups. Similar proportions of journal, academic, government, and parliamentary publications were cited by both actor groups. Compared to non-industry actors, industry actors cited significantly higher proportions of publications by alcohol industry-linked organisations (15% industry vs 3% non-industry, $p < .0001$) and self-regulatory bodies (20% industry vs 12% non-

**Table 2. Submission characteristics.**

| | | Industry actor submissions | | | Non-industry actor submissions | | |
|---|---|---|---|---|---|---|---|
| | | ANPHA (n = 20) | NSW (n = 25) | ANPHA and NSW (n = 45) | ANPHA (n = 10) | NSW (n = 16) | ANPHA and NSW (n = 26) |
| Main body | Range page length | 1–41 | 1–28 | 1–41 | 1–39 | 2–24 | 1–39 |
| | Median page length | 9 | 4 | 6 | 15 | 4.5 | 7 |
| | Mean page length | 11 | 7 | 9 | 17 | 7 | 11 |
| Appendices | No. of submissions with appendices | 5 | 3 | 8 | 3 | 4 | 7 |
| | Range page length | 1–136 | 3–10 | 1–136 | 6–56 | 4–34 | 4–56 |
| | Mean page length (per submission with appendices) | 32 | 6 | 22 | 25 | 18 | 20 |
| Cited evidence | Total | 261 | 259 | 520 | 384 | 346 | 730 |
| | Range per submission | 0–42 | 0–110 | 0–110 | 2–112 | 2–66 | 2–112 |
| | Median per submission | 11 | 2 | 4 | 37 | 16 | 18 |
| | Mean per submission | 13 | 10 | 12 | 38 | 22 | 28 |

Note. ANPHA: Australian National Preventive Health Agency issues paper Alcohol advertising: The effectiveness of current regulatory codes in addressing community concerns. NSW: New South Wales Alcoholic Beverages Advertising Prohibition Bill.

industry, $p < .0001$) and correspondingly fewer publications by organisations not linked to the alcohol industry (8% industry vs 17% non-industry, $p < .0001$).

Industry actors were significantly more likely to cite older evidence (published more than 10 years before the consultation) than non-industry actors. This applied both to evidence cited overall (32% industry vs 9% non-industry, $p < .0001$) and to peer-reviewed journal and academic publications specifically (57% industry vs 13% non-industry, $p < .0001$).

Small proportions of journal and academic publications cited by each actor group were identified as alcohol industry-funded or -linked (9% industry vs 4% non-industry, $p = .0139$). Many cited publications (n = 166) did not provide a clear funding or disclosure statement, which prevented assessment of independence for a substantial proportion of journal and academic publications (44% industry vs 24% non-industry, $p < .0001$).

Potential outlier scores were identified. When analyses were re-run with outlier scores removed, no outlier effects were detected.

As shown in Table 4, proportions of evidence cited by industry actors that featured any two (54%) or all three (14%) indicators of higher quality were significantly and substantially smaller than for non-industry actors (80%, $p < .0001$ and 37%, $p < .0001$ respectively). The proportions reduced for both actor groups when high relevance was considered in addition to the indicators of higher quality; however, the marked difference remained between evidence cited by industry and non-industry actors. Only 27% (141 of 520) of evidence cited by industry actors was highly relevant and featured at least two indicators of higher quality, compared to 58% (420 of 730) of evidence cited by non-industry actors.

## Discussion

### Principal findings

To our knowledge, this is the first time evidence put forward in alcohol policy submissions for consideration by decision makers has been assessed for relevance and quality. Evidence cited by industry actors was less likely to be highly relevant or to feature indicators of higher quality than evidence cited by non-industry actors. Overall, only 27% of evidence cited by industry actors compared to 58% of evidence cited by non-industry actors was about alcohol

**Table 3. Comparisons of subject matter relevance, publication type, time since publication, and independence.**

| | | Evidence cited in industry actor submissions (n = 520) | Evidence cited in non-industry actor submissions (n = 730) | Chi-square statistic | p-value | Phi coefficient |
|---|---|---|---|---|---|---|
| | | n (%) | n (%) | | | |
| Subject matter relevance | Alcohol advertising (High) | 262 (50) | 531 (73) | 65.90 | < .0001* | .23 |
| | Alcohol but not advertising (Medium) | 145 (28) | 125 (17) | 37.82 [a] | < .0001* | .17 |
| | Advertising but not alcohol (Medium) | 49 (9) | 32 (4) | - | - | - |
| | Unrelated to alcohol or advertising (Low) | 13 (3) | 22 (3) | 0.29 | .5902 | .02 |
| | Unknown subject matter | 51 (10) | 20 (3) | - | - | - |
| Publication type | Peer-reviewed journal | 165 (32) | 276 (38) | 7.48 [b] | .0062 | .08 |
| | Academic publication | 29 (6) | 52 (7) | - | - | - |
| | Government publication | 85 (16) | 131 (18) | 0.52 | .4708 | .02 |
| | Parliamentary publication | 11 (2) | 15 (2) | 0.01 | .9203 | .00 |
| | Publication by an alcohol industry-linked organisation | 76 (15) | 21 (3) | 58.63 | < .0001* | .22 |
| | Publication by an organisation not linked to the alcohol industry | 39 (8) | 126 (17) | 25.13 | < .0001* | .14 |
| | Publication from an alcohol advertising self-regulatory body | 102 (20) | 85 (12) | 15.29 | < .0001* | .11 |
| | Media coverage | 2 (<1) | 20 (3) | 9.72 | .0018* | .09 |
| | Unknown publication type | 11 (2) | 4 (<1) | - | - | - |
| Time since publication | Published more than 10 years before the consultation—all evidence [c] | 137 (32) | 59 (9) | 97.80 | < .0001* | .30 |
| | Published within 10 years of the consultation—all evidence [c] | 298 (68) | 630 (91) | - | - | - |
| | Published more than 10 years before the consultation—journal/academic [d] | 110 (57) | 44 (13) | 111.39 | < .0001* | .46 |
| | Published within 10 years of the consultation—journal/academic [d] | 83 (43) | 283 (87) | - | - | - |
| | Unknown publication year | 85 (16) | 41 (6) | - | - | - |
| Independence [e] | Independent | 91 (47) | 234 (71) | 31.42 | < .0001* | .25 |
| | No apparent alcohol industry connection | 86 (44) | 80 (24) | 21.26 | < .0001* | .20 |
| | Alcohol industry-funded | 7 (4) | 3 (<1) | 6.04 [f] | .0139 | .11 |
| | Alcohol industry-linked | 10 (5) | 9 (3) | - | - | - |

Note.

* Significant at Bonferroni-adjusted p value <0.0026.

[a]Medium relevance categories were combined for the chi-square comparison.

[b]Peer-reviewed journal and academic publications were combined for the chi-square comparison.

[c]Count includes all cited evidence, excluding those with unknown publication year.

[d]Count includes only peer-reviewed journal and academic publications, excluding those with unknown publication year.

[e]Independence is reported for peer-reviewed journal and academic publications only.

[f]Alcohol industry-funded and alcohol industry-linked categories were combined for the chi-square comparison.

advertising (i.e., highly relevant) and displayed two or more indicators of higher quality: i) published in a peer-reviewed journal or academic source, ii) independent of or with no apparent connection to the alcohol industry, and/or iii) published within 10 years of the consultation.

**Table 4. Proportions of evidence with indicators of higher quality and relevance.**

| Indicators of higher quality and relevance | Evidence cited in industry actor submissions | Evidence cited in non-industry actor submissions | Chi-square statistic | p-value | Phi coefficient |
|---|---|---|---|---|---|
| | n (% of 520) | n (% of 730) | | | |
| High relevance [a] | 262 (50) | 531 (73) | 65.90 | < .0001* | .23 |
| Any two (or more) indicators of higher quality [b] | 283 (54) | 582 (80) | 91.22 | < .0001* | .27 |
| High relevance plus any two (or more) indicators of higher quality | 141 (27) | 420 (58) | 113.59 | < .0001* | .30 |
| All three indicators of higher quality [b] | 74 (14) | 271 (37) | 79.64 | < .0001* | .25 |
| High relevance plus all three indicators of higher quality | 38 (7) | 214 (29) | 91.38 | < .0001* | .27 |

Note.

* Significant at Bonferroni-adjusted p value <0.0026.

[a]Quality was not considered.

[b]Subject matter relevance was not considered.

## Comparison with other studies

A distinct polarisation of views on potential regulatory action was identified, with industry actors opposing and non-industry actors supporting further regulation of alcohol advertising. This finding is consistent with Allen and Clarke's analysis of submissions to the New Zealand Ministerial Forum on Alcohol Advertising and Sponsorship [16]. In the present study, the substantial participation of advertising and media organisations (accounting for 32% of analysed submissions), alongside major sporting bodies (3% of submissions), reflects the position of these industries as close allies of the alcohol industry in resisting advertising controls [8, 49]. This has previously been observed in the active roles of these industries in resisting tobacco advertising bans [40, 50].

The findings demonstrate several strategies known to be used by the alcohol industry to undermine the evidence base for strengthened policy approaches. First, the strategy of omitting evidence [15] was demonstrated by the substantial number of industry actor submissions that expressed opposition to further regulation of alcohol advertising but provided no supporting evidence. Second, the selective citation of favourable evidence [15] can be seen in industry actors' substantial use of industry-linked publications and dated sources of evidence. Industry actors may have identified older evidence as more favourable to their position, as opposed to the more contemporary evidence cited by non-industry actors in support of further regulation of alcohol advertising. Third, industry actors appear to have sought to change the evidential landscape within which the alcohol advertising policy debate was conducted [8, 51] via their greater reliance on less relevant evidence, encouraging attention on other aspects of alcohol policy debates rather than more focused attention on alcohol advertising policy. As such, claims by industry actors that the evidence base is insufficient to support proposals to further regulate alcohol advertising [15, 39] appear to relate more to the industry's selectivity in acknowledging evidence than any genuine deficiency in the evidence base. The exclusion of industry actors from alcohol policy development processes may be necessary if these established industry strategies are to be avoided [8, 12–15].

While the present study is not directly comparable with Hatchard et al.'s analysis of evidence cited in tobacco industry submissions, similar patterns emerged [17]. A substantial proportion of evidence cited by industry actors in both studies was focused on 'parallel' subject matter (i.e., of medium or low relevance) rather than being directly relevant to the consultation

topic. About 50% of industry actor evidence in the present study and 34% of tobacco industry evidence in Hatchard et al. was of high relevance. In both studies, significantly lower proportions of evidence cited by industry actors were highly relevant and featured indicators of higher quality compared to the points of comparison (i.e., systematic review evidence in Hatchard et al. and evidence cited by non-industry actors in the present study).

## Implications for policymakers

In both consultations that were the focus of this study, policymakers received a greater number of submissions from industry actors than non-industry actors. The volume of industry actor opposition to increased regulation of alcohol advertising may have been noticeable to policymakers, tempering their willingness to change the policy status quo. Policymakers who act to restrict alcohol advertising would need to be prepared to tolerate opposition from industry actors but can expect support from non-industry actors based on a solid foundation of evidence.

Industry actor submissions in the present study commonly cited sources published more than 10 years before the consultation. While not necessarily obsolete or irrelevant, as Cullen et al. identified, dated sources of evidence are unlikely to provide up-to-date support for the claims made in submissions [5]. Policymakers were presented with evidence that was less likely to be relevant to the contemporary alcohol advertising environment, given the evolution and proliferation of media channels via which audiences are exposed to alcohol advertising [45]. Older research is also less likely to have incorporated rapid advances in research methodologies, such as improved tools for measuring advertising exposure [46–48]. Policy consultations may benefit from the introduction of quality requirements for submitted evidence [19], which include regard for the timeliness of evidence.

While the age of evidence can be readily assessed by policymakers, assessing independence from vested interests may be more difficult [19, 52], as was found in the present study. As journals adopt more comprehensive disclosure requirements [52–54], conflicts should be more visible in future publications, but this would still require policymakers to consult the cited evidence. To further aid the assessment of independence in policy consultations, it may be useful for the administering agency to define potential conflicts of interests and require submitters to declare any potential conflicts relevant to the evidence they present [17–19].

Action to better control alcohol advertising did not appear to follow either of the consultations included in this study. The consultation reports suggested that evidence presented by non-industry actors was convincing, but was not able to overcome representations of industry actors. The committees found the existing self-regulatory regime to be "inadequate" (p. 6) [55] and "insufficient and failing" (p. 86) [42] to protect young people from alcohol advertising. However, ANPHA was disbanded by the Australian Government in the same year the final report was published and its recommendations to strengthen regulations were not progressed. The NSW Parliamentary committee considering the Alcohol Beverages Advertising Prohibition Bill 2015 recommended the Bill not be passed. Despite having rejected some of the "sophisticated arguments" put forward by the alcohol and advertising industries, such as "why there is no causal link between advertising and consumption" (p. 33) [42], other industry arguments appeared to create room for doubt among committee members. For example, an alcohol producer presented evidence about health benefits of moderate alcohol use. While the committee acknowledged "compelling evidence that alcohol in fact poses no real health benefits" (p. 33) [42], rather than consider the issue settled, the committee recommended that the NSW Health Department "closely examine the issue of whether there is any safe level of alcohol consumption" and that "this research should determine whether alcohol advertising should have

further restrictions applied to it" (p. 34) [42]. The committee also appeared to accept the concerns of industry actors that "the Bill will have a detrimental impact on the alcohol and advertising industries and will lead to a range of unintended consequences" (p. 56) [42], such as making it difficult for new market entrants and small businesses to grow.

### Strengths and limitations of the study

The primary strength of this study was the inclusion of all accessible submissions made to two consultations on alcohol advertising regulation, capturing a range of policy actors who participated in the consultations and enabling comparisons between actor groups. The inclusion of multiple consultations strengthened our ability to consider the implications of findings for improving the use of evidence in policy processes.

A limitation was that reliance on publicly available submissions precluded access to confidential submissions; however, there appeared to have been only three confidential submissions that could not be accessed. Second, consistent with difficulties experienced by those examining industry submissions to consultations in other policy contexts (e.g., ultra-processed foods and sugar-sweetened beverages [19, 20]), the referencing practices within submissions impacted the researchers' ability to identify cited evidence. These practices varied between submissions as there was no requirement to reference evidence in a manner that would allow others to locate it or even to reference evidence at all. Third, the approach of the present study did not give an indication of the weight of the use of different sources of evidence. Regardless of whether an item of evidence was cited once or 10 times within a submission, it was recorded once for that submission. Fourth, independence for journal and other academic publications was assessed only on the basis of disclosures within publications; relevant information from other sources may have been missed. Almost one-third of journal and academic publications cited by submissions did not include a statement from which independence could reasonably be assessed. In some of these cases, disclosure statements were made but they were ambiguous (e.g., "the usual disclaimers apply") and therefore of limited value to readers wishing to assess potential conflicts. A fifth limitation was that the passing of time since the consultations may have reduced the accessibility of cited evidence in a small number of instances (S2 Table). Finally, as this study focussed on a single policy issue, generalizability of results to other alcohol policy issues is unknown.

### Unanswered questions and future research

There may be value in replicating this study using consultations that explore other alcohol policy topics (e.g., price or availability controls) to determine whether similar patterns exist. Future assessments of the relevance and quality of evidence may consider weighting the criteria to prioritise the indicators of greatest importance to the issue under consideration. While analyses of submissions provide useful insights into the information collected to inform policymaker deliberations, they are not able to tell us how policymakers assess information they receive. Building a better understanding of how policymakers evaluate and apply evidence would help to identify practical options for improving the uptake of research evidence in policy deliberations.

### Conclusions

The value of the evidentiary contribution of industry actors to consultations on alcohol advertising policy appears to be limited. Modifications to consultation processes, such as exclusion of industry actors, quality requirements for submitted evidence, minimum standards for

referencing evidence, and requirements to declare potential conflicts, may improve the transparency and application of evidence and public health outcomes of policy consultations.

## Supporting information

**S1 Table. Number of submissions by actor type.**
(DOCX)

**S2 Table. Accessibility of cited evidence by actor group, consultation and publication type.**
(DOCX)

## Author Contributions

**Conceptualization:** Julia Stafford, Simone Pettigrew.

**Formal analysis:** Julia Stafford.

**Investigation:** Julia Stafford, Hannah Pierce.

**Methodology:** Julia Stafford, Tanya Chikritzhs, Simone Pettigrew.

**Project administration:** Julia Stafford.

**Supervision:** Tanya Chikritzhs, Simone Pettigrew.

**Writing – original draft:** Julia Stafford.

**Writing – review & editing:** Tanya Chikritzhs, Hannah Pierce, Simone Pettigrew.

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
