## [Decision Letter · Decision Letter 0]

14 Oct 2021

PONE-D-21-26197An evaluation of the evidence submitted to Australian alcohol advertising policy consultationsPLOS ONE

Dear Dr. Stafford,

Thank you for submitting your manuscript to PLOS ONE. After careful consideration, we feel that it has merit but does not fully meet PLOS ONE’s publication criteria as it currently stands. Therefore, we invite you to submit a revised version of the manuscript that addresses the points raised during the review process.

The reviewers were highly positive about the value and rigour of this manuscript and make some minor suggestions to strengthen its presentation. Please address each of these comments. I also concur with Reviewer 1 that the material under Acknowledgments is more appropriately located within a competing interests statement - I ask that you review the journal's policy on declaring competing interests to decide what is most appropriately declared. I do not necessarily think this needs to be included in the limitations section, but you might include a paragraph on 'reflexivity' if you think it important to document how the team's expertise and social location shaped the work and in what ways. I look forward to receiving your revised manuscript.

We look forward to receiving your revised manuscript.

Kind regards,

Quinn Grundy, PhD, RN

Academic Editor

PLOS ONE

Journal Requirements:

a) Did participants provide their written or verbal informed consent to participate in this study?

3. Peer review at PLOS ONE is not double-blinded (https://journals.plos.org/plosone/s/editorial-and-peer-review-process). For this reason, authors should include in the revised manuscript all the information removed for blind review

Reviewers' comments:

Reviewer's Responses to Questions

**Comments to the Author**

1. Is the manuscript technically sound, and do the data support the conclusions?

Reviewer #1: Yes

Reviewer #2: Yes

2. Has the statistical analysis been performed appropriately and rigorously? 

Reviewer #1: Yes

Reviewer #2: Yes

3. Have the authors made all data underlying the findings in their manuscript fully available?

Reviewer #1: Yes

Reviewer #2: Yes

4. Is the manuscript presented in an intelligible fashion and written in standard English?

Reviewer #1: Yes

Reviewer #2: Yes

5. Review Comments to the Author

Reviewer #1: An evaluation of the evidence submitted to Australian alcohol advertising policy consultations

Overview:

This is a very well designed, explained and executed piece of research that will no doubt contribute beneficially and insightfully to the field.

The research has been rigorous and is well contextualised and explained in relation to relevant literature, in particular on alcohol and secondarily, tobacco.

Quite often, questions that did arise were subsequently addressed in the later discussion.

The findings have yielded significant descriptive statistics that help substantiate the conclusions.

The comments below are minor and there were no issues found of any substance that would undermine a very positive evaluation of the standard and significance of the research. The authors are to be commended.

The Abstract is very good and gives a succinct summary of the research

I would recommend adding to the introductory Abstract paragraph ‘Evaluation of “The value of the evidentiary contribution of industry actors to consultations on alcohol

advertising policy appears to be limited”

Body of the article

p.8 of the MS pdf.

Re: ‘ls of cited evidence were extracted and coded against a framework adapted from previous research (primary codes: subject matter relevance, type of publication, time since publication, and independence from industry)’.

Q might the authors consider weighting the criteria with independence from industry the most highly weighted?

p. 12 of the MS pdf.

Line 115-16

Should say investigations by researchers [ie in order to differentiate this from governments]

method

p. 13 of the MS pdf.

Q why would human subject ethics permission be sought to examine publicly available submissions?

Perhaps other aspects of the research should be mentioned in this regard eg. Interviews or whatever necessitated human subject research.

p. 13 of the MS pdf.

the two inquiries that constitute the source of data are well described. However the authors could say why it was important. Eg was it to compare the two as justification that the results were not ideosyncratic to one inquiry? Was it to include both national and state inquiries etc.?

p.16 of the MS pdf.

Most of the codings are well explained and where relevant, the source in other relevant research. Regarding time since publication defined as up to 10 years or more than 10 years. Its there a rationale for this coding? Also, elsewhere in the text 15 years was mentioned. How does this fit the coding? [Also described on p.18 of the MS].

p. 19 MS pdf.

Line 276

‘Where detected, analyses were re-run with extreme scores removed to determine whether results differed.’

Q. isn’t it stated later that there were no cases of extreme scores?

p. 27 of MS pdf.

Principle findings brings together the most significant findings of the research.

The authors give a very good succinct summation of the main findings.

Final para

‘Modifications to consultation processes, such as exclusion of industry actors, quality requirements for submitted evidence, minimum standards for referencing evidence, and requirements to declare potential conflicts, may improve the transparency and application of evidence and public health outcomes of policy consultations’.

These are very reasonable recommendations that come out of this work with application more broadly to government inquiries. If submissions draw on evidence that is no publicly traceable for inspection/accuracy/bias, then this is a shortcoming that could seemingly be risk managed along lines suggested.

Under acknowledgments, this reviewer is not sure of the relevance of the declarations that presumably apply to different authors’ contributions to submissions that would have been included in the inquiry. Acknowledgments usually refer to funding sources and the sort of statements made here should perhaps go in potential limitations and how/why bias is/not introduced and also discuss whether the reported research in this article was designed after or contemporaneously with these organisational submissions in which authors were involved.

Alternatively, some of this may belong in declarations of interest. Since authors have contributed to some of the submissions to the inquires under analysis and in one case, to the final report. This does not mean to say the person should not be an author but to explain why these contributions do/not bias the research.

Reviewer #2: Many thanks for the opportunity to review this well written and important manuscript. The authors have conducted a thorough analysis of responses to two public consultations focusing on alcohol advertising policy using previously published methods. In this way the study is an important contribution to the growing body of literature on industry use and representation of evidence in the consultation process and adds a novel dimension by analysing and comparing industry with non-industry submissions.

I have only minor comments for the authors to consider addressing:

Page 5 – paragraph from lines 91-103 – the authors could extend this further by making reference to additional studies that show that similar tactics are employed by the ultra-process food industry and the sugar sweetend beverage industry e.g.

Fooks GJ, Williams S, Box G, Sacks G. Corporations’ use and misuse of evidence to influence health policy: a case study of sugar-sweetened beverage taxation. Globalization and Health 2019; 15(1): 56.

Lauber K, McGee D, Gilmore AB. Commercial use of evidence in public health policy: a critical assessment of food industry submissions to global-level consultations on non-communicable disease prevention. BMJ Global Health 2021; 6(8): e006176.

This could help to further demonstrate the cross-industry nature of these tactics.

Page 8 lines 167-172 – the authors describe who was classified as industry – where their other commercial actors e.g. those from the tourism/entertainment industry or bars/clubs (where these considered “retailers”) and where these included as industry or non-industry? This may be helpful in thinking though some of the wider partnership and allies that can be built by the industry and whose interests are served or threatened by different policy outcomes as the authors have shown for the advertising industry.

Page 22: “In both studies, significantly lower proportions of evidence cited by industry actors were highly relevant and featured indicators of higher quality compared to the points of comparison (i.e., systematic review evidence in Hatchard et al. and evidence cited by non-industry actors in the present study)”

The authors may want to add some addition text to clarify that industry submissions have lower levels of features indicating high quality.

Discussion/conclusion – the authors make several well-considered recommendations on how the consultation process may be strengthened. They may wish to highlight to the reader that similar recommendations have been made in other studies in the context of other policies and industries, which may strengthen the case for calling for such adjustments given other research groups in other locations and time points have also demonstrated the need for such measures if policy-making is to serve the public interest and draw on the best available evidence.

6. PLOS authors have the option to publish the peer review history of their article (what does this mean?). If published, this will include your full peer review and any attached files.

Reviewer #1: **Yes: **Prof Linda Hancock

Reviewer #2: No

---

## [Author Response · Author response to Decision Letter 0]

20 Nov 2021

Please see attached Response to Reviewers.

---

## [Editor Report · Decision Letter 1]

29 Nov 2021

An evaluation of the evidence submitted to Australian alcohol advertising policy consultations

PONE-D-21-26197R1

Dear Dr. Stafford,

We’re pleased to inform you that your manuscript has been judged scientifically suitable for publication and will be formally accepted for publication once it meets all outstanding technical requirements.

Kind regards,

Quinn Grundy, PhD, RN

Academic Editor

PLOS ONE
---

## [Editor Report · Acceptance letter]

3 Dec 2021

PONE-D-21-26197R1 

An evaluation of the evidence submitted to Australian alcohol advertising policy consultations  

Dear Dr. Stafford:

I'm pleased to inform you that your manuscript has been deemed suitable for publication in PLOS ONE. Congratulations! Your manuscript is now with our production department. 

Kind regards, 

on behalf of

Dr. Quinn Grundy 

Academic Editor

PLOS ONE